# Diffusion-based Negative Sampling on Graphs for Link Prediction

Anonymous

## ABSTRACT

Link prediction is a fundamental task for graph analysis with important applications on the Web, such as social network analysis and recommendation systems, *etc.* Modern graph link prediction methods often employ a contrastive approach to learn robust node representations, where negative sampling is pivotal. Typical negative sampling methods aim to retrieve hard examples based on either predefined heuristics or automatic adversarial approaches, which might be inflexible or difficult to control. Furthermore, in the context of link prediction, most previous methods sample negative nodes from existing substructures of the graph, missing out on potentially more optimal samples in the latent space. To address these issues, we investigate a novel strategy of *multi-level* negative sampling that enables negative node generation with flexible and controllable "hardness" levels from the latent space. Our method, called Conditional Diffusion-based Multi-level Negative Sampling (DMNS), leverages the Markov chain property of diffusion models to generate negative nodes in multiple levels of variable hardness and reconcile them for effective graph link prediction. We further demonstrate that DMNS follows the sub-linear positivity principle for robust negative sampling. Extensive experiments on several benchmark datasets demonstrate the effectiveness of DMNS.

**ACM Reference Format:**
Anonymous. 2018. Diffusion-based Negative Sampling on Graphs for Link Prediction. In *Proceedings of Make sure to enter the correct conference title from your rights confirmation emai (Conference acronym 'XX).* ACM, New York, NY, USA, 10 pages. https://doi.org/XXXXXXX.XXXXXXX

## 1 INTRODUCTION

Graph, which consists of nodes and links between them, is a ubiquitous data structure for real-world networks and systems. Link prediction [27] is a fundamental problem in graph analysis, aiming to model the probability that two nodes relate to each other in a network or system. Alternatively, given a query node, link prediction aims to rank other nodes based on their probability of linking to the query node. Graph link prediction enables a wide range of applications on the Web, such as friend suggestions in social networks [6], products recommendation in e-commerce platforms [48], and knowledge graph completion [1] for many Web-scale knowledge bases.

A prominent approach that has been extensively studied for link prediction is graph representation learning. It trains an encoder to produce low-dimensional node embeddings that capture the original graph topology in a latent space. Toward link prediction, many graph representation learning methods [13, 41, 47] follow the noise contrastive estimation approach [12], which resorts to sampling a set of positive and negative nodes for any query node. Specifically, the encoder is trained to capture graph topology by bringing positive node pairs closer while separating the negative pairs in the embedding space. While sampling the positive examples for link prediction is relatively straightforward (*e.g.*, one-hop neighbors of the query node), sampling negative examples involves a huge search space that is quadratic in the number of nodes and a significant fraction of unlinked node pairs could be false negatives since not all links may be observed. Hence, studying negative sampling for contrastive link prediction on graphs is a crucial research problem.

Many strategies have been proposed for negative sampling on graphs, yet it is often challenging to flexibly model and control the quality of negative nodes. While uniform negative sampling [13, 41] is simple, it ignores the quality of negative examples: Harder negative examples can often contribute more to model training than easier ones. Many studies explore predefined heuristics to select hard negative nodes, such as popularity [29], dynamic selections based on current predictions [51], $k$-hop neighborhoods [1], Personalized PageRank [48], *etc.* However, heuristics not only need elaborate designs, but also tend to be inflexible and may not extend to different kinds of graph. For instance, homophilous and heterophilous graphs exhibit different connectivity patterns, which means a good heuristic for one would not work well for the other. Besides heuristics, automatic methods leveraging generative adversarial networks (GANs) [4, 43] are also popular. They aim to learn the underlying distribution of the nodes and retrieve harder ones automatically. However, it is still difficult to flexibly control the "hardness" of the negative examples for more optimal contrastive learning. Furthermore, it has been shown that the hardest negative examples may impair the performance [9, 44], when they are nearly indistinguishable from the positive ones particularly in the early phase of training. To overcome this, we propose a strategy of *multi-level* negative sampling, where we can flexibly control the hardness level of the negative examples according to the need. For instance, easier negative examples can be used to warm-up model training, while harder ones are more critical to refining the decision boundary. Overall, a well-controlled mixture of easy and hard examples are expected to improve learning.

The idea of multi-level negative sampling immediately brings up the second question: Where do we find enough negative examples of variable hardness? Most negative sampling approaches [1, 13, 29, 41, 48, 51] are limited to sampling nodes from the observed graph. However, observed graphs in real-world scenarios are

often noisy or incomplete, which are not ideal sources for directly sampling a sufficient mixture of negative examples at different levels of hardness. To tackle this issue, we propose to synthesize more negative nodes to complement the real ones in the latent space. The latent space can potentially provide infinite negative examples with arbitrary hardness, to aid the generation of multi-level negative examples in a flexible and controllable manner. Although a few studies [18, 23] also utilize GANs to generate additional samples in the latent space, they are not designed to synthesize multi-level examples.

To materialize our multi-level strategy, a natural choice is diffusion models. Recently, diffusion models have achieved promising results in generation tasks [15, 38], particularly in visual applications [7, 16]. While GAN-based models are successful in generating high-fidelity examples, they face many issues in training such as vanishing gradients and mode collapse [2, 3]. In contrast, diffusion models follow a reconstruction mechanism that offers a stable training process [28]. More specific to our context, a desirable property of diffusion model is that it utilizes a Markov chain with multiple time steps to denoise random input, where the sample generated at each time step is conditioned on the sample in the immediately preceding step. Hence, during the generation process, we can naturally access the generated samples at different denoised time steps to achieve our multi-level strategy. Hence, we propose a novel diffusion-based framework to generate negative examples for graph link prediction, named Conditional Diffusion-based Multi-level Negative Sampling (DMNS). On one hand, we employ a diffusion model to learn the one-hop connectivity distribution of any given query node, *i.e.*, the positive distribution in link prediction. The diffusion model allows us to flexibly control the hardness of each negative example by looking up a specified time step $t$, ranging from a virtually positive example (or indistinguishable from the positives) when $t = 0$ and harder examples as $t \rightarrow 0$, to easier ones amounting to random noises as $t \rightarrow \infty$. On the other hand, we adopt a conditional diffusion model [7, 16], which is designed to explicitly condition the generation on side information. In graph link prediction, we condition the positive distribution of a specific query node on the node itself, and thus obtain query-specific diffusion models. Theoretically, we show that the density function of our negative examples obeys the sub-linear positivity principle [47] under some constraint, ensuring robust negative sampling on graphs.

Our contributions in this work are summarized as follows. 1) We investigate the strategy of multi-level negative sampling on graphs for contrastive link prediction. 2) We propose a novel framework DMNS based on a conditional diffusion model, which generates multi-level negative examples that can be flexibly controlled to improve training. 3) We show that the distribution function of our negative examples follows the sub-linear positivity principle under a defined constraint. 4) We conduct extensive experiments on several benchmark datasets, showing that our model outperforms state-of-the-art baselines on graph link prediction.

## 2 RELATED WORK

*Link prediction.* The success of deep learning has motivated extensive studies on graph representation learning [13, 22, 42]. On learned graph representations, a simple strategy for link prediction is to employ a node-pair decoder [11]. More sophisticated approach exploits the representations of the enclosing subgraphs. SEAL [50] proposes the usage of local subgraphs based on the $\gamma$-decaying heuristic theory. ScaLED [26] utilizes random walks to efficiently sample subgraphs for large-scale networks. BUDDY [5] proposes subgraph sketches to approximate essential features of the subgraphs without constructing explicit ones.

*Negative sampling.* Graph representation learning approaches for link prediction commonly employ contrastive strategies [48, 49]. The contrastive methods require effective negative sampling to train the graph encoders. Various negative sampling heuristics have been proposed, such as based on the popularity of examples [29], current prediction scores [51] and selecting high-variance samples [8]. On graphs, SANS [1] select negative examples from $k$-hop neighborhoods. MixGCF [19] synthesizes hard negative examples by hop and positive mixing. MCNS [47] develops a Metropolis-Hasting sampling approach based on the proposed sub-linear positivity theory. Another line of research utilizes generative adversarial networks (GANs) for automatic negative sampling. Among them, GraphGAN [43] and KBGAN [4] learns a distribution over negative candidates, while others generate new examples not found in the original graph [18, 23, 24]. However, these methods cannot easily control the hardness of the negative examples, which is the key motivation of our multi-level strategy.

*Diffusion models.* Diffusion models [15, 37, 38, 40] have become state-of-the-art generative models, which gradually inject noises into the data and then learn to reverse this process for sampling. Denoising diffusion probabilistic models (DDPMs) [15, 37] aims to predict noises from the diffused output at arbitrary time steps. Score-based Generative Models (SGMs) [38–40] aims to predict the score function $\nabla \log(p_{x_t})$. They have different approaches but are shown to be equivalent to optimizing a diffusion model. Conditional diffusion models [7, 16] permit explicit control of generated samples via additional conditions, which enables a wide range of applications such as visual generation [35], NLP [10], multi-modal generation [14, 30], *etc*. Recently, some studies have adopted diffusion models in graph generation tasks. EDM [17] learns a diffusion process to work on 3d molecule generation, while GDSS [20] learns to generate both node features and adjacency matrices. However, they do not aim to generate multi-level node samples for graph link prediction. We refer readers to a comprehensive survey [45].

## 3 PRELIMINARIES

In this section, we briefly introduce the background of graph neural networks and diffusion models, which are the foundation of our proposed DMNS.

*Graph neural networks.* Message-passing GNNs usually resort to multi-layer neighborhood aggregation, in which each node recursively aggregates information from its neighbors. Specifically, the representation of node $v$ in the $l$-th layer, $\mathbf{h}_v^l \in \mathbb{R}^{d_h^l}$, is constructed as

$$\mathbf{h}_v^l = \sigma \left( \text{AGGR}(\mathbf{h}_v^{l-1}, \{\mathbf{h}_i^{l-1} : i \in \mathcal{N}_v\}; \omega^l) \right), \quad (1)$$

where $d_h^l$ is the dimension of node representations in the $l$-th layer, Aggr$(\cdot)$ denotes an aggregation function such as mean-pooling [22], self-attention [42] or concatenation [13], $\sigma$ is an activation function, $\mathcal{N}_v$ denotes the set of neighbors of $v$, and $\omega^l$ denotes the learnable parameters in layer $l$.

*Diffusion models.* Denoising diffusion model (DDPM) [15] boils down to learning the Gaussian transitions of Markov chains. Specifically, DDPM consists of two Markov chains: a forward diffusion process, and a backward denoising process (also known as the "reverse" process). The forward chain transforms input data to complete noise by gradually adding Gaussian noise at each step:

$$q(\mathbf{x}_{1:T}|\mathbf{x}_0) = \prod_{t=1}^{T} q(\mathbf{x}_t|\mathbf{x}_{t-1}), \tag{2}$$

$$q(\mathbf{x}_t|\mathbf{x}_{t-1}) = \mathcal{N}(\mathbf{x}_t; \sqrt{1-\beta_t}\mathbf{x}_{t-1}, \beta_t\mathbf{I}), \tag{3}$$

where $T$ is the total number of time steps of the diffusion process, and $\beta_t$ denotes the variances in the diffusion process, which can be learnable or fixed constants via some scheduling strategy. By a reparameterization trick, the closed form of output $\mathbf{x}_t$ at arbitrary time step $t$ can be obtained as

$$\mathbf{x}_t = \sqrt{\bar{\alpha}_t}\mathbf{x}_0 + \sqrt{1-\bar{\alpha}_t}\epsilon_t, \tag{4}$$

where $\alpha_t = 1 - \beta_t$, $\bar{\alpha}_t = \prod_{i=1}^{t} \alpha_i$, and $\epsilon_t \sim \mathcal{N}(\mathbf{0}, \mathbf{I})$. The backward denoising process learns to reconstruct the input from noise:

$$p_\theta(\mathbf{x}_{0:T}) = p(\mathbf{x}_T) \prod_{t=1}^{T} p_\theta(\mathbf{x}_{t-1}|\mathbf{x}_t) \tag{5}$$

$$p_\theta(\mathbf{x}_{t-1}|\mathbf{x}_t) = \mathcal{N}(\mathbf{x}_{t-1}; \mu_\theta(\mathbf{x}_t, t), \Sigma_\theta(\mathbf{x}_t, t)) \tag{6}$$

$$p(\mathbf{x}_T) = \mathcal{N}(\mathbf{x}_T; \mathbf{0}, \mathbf{I}), \tag{7}$$

where $\theta$ parameterizes the diffusion model. To model the joint distribution $p_\theta(\mathbf{x}_{t-1}|\mathbf{x}_t)$, DDPM sets $\Sigma_\theta(\mathbf{x}_t, t) = \beta_t\mathbf{I}$ as scheduled constants, and derive $\mu_\theta(\mathbf{x}_t, t)$ by a reparameterization trick as

$$\mu_\theta(\mathbf{x}_t, t) = \frac{1}{\sqrt{\alpha_t}}\mathbf{x}_t - \frac{1-\alpha_t}{\sqrt{\alpha_t}\sqrt{1-\bar{\alpha}_t}}\epsilon_\theta(\mathbf{x}_t, t), \tag{8}$$

where $\epsilon_\theta(\mathbf{x}_t, t)$ is a function to estimate the source noise $\epsilon$ and can be implemented as a neural network. The objective now learns to predict added noises instead of means, which has been shown to achieve better performance.

## 4 PROPOSED MODEL: DMNS

In this section, we introduce the proposed method DMNS to generate multi-level negative examples for graph link prediction. Before we begin, we first present the overall framework of DMNS in Fig. 1. We employ a standard GNN encoder to obtain node embeddings, which captures content and structural neighborhood information. Next, we train a diffusion model to learn the neighborhood distribution conditioned on the query node. From the model we sample several output embeddings at different time steps, to serve as negative examples at multi-level hardness for contrastive learning.

### 4.1 Multi-level Negative Sample Generation

Given a query node, we aim to learn the distribution of its 1-hop neighbors, *i.e.*, its positive distribution for the link prediction task. Vanilla diffusion model only generates generic examples without the ability to personalize for the query node. To this end, we leverage the conditional diffusion model [7, 16], taking query node embeddings as additional information for sample generation. From

that, we can perform multi-level negative sampling by extracting generated node embeddings from multiple time steps. The choice of time steps empowers us to control the hardness of negative examples. For instance, the embedding output from the final step of the denoise (reverse) process ($t = 0$) is the hardest to distinguish from the positive nodes in the latent space, while its counterparts from earlier steps (larger $t$'s) can be taken as progressively easier negative nodes. In general, the time step $t$ is a proxy to the hardness of negative examples: a smaller $t$ gives harder examples. Thus, we can automatically incorporate such multi-level negative examples into training the link prediction task.

Specifically, for a query node $v$, we obtain its embedding $\mathbf{h}_v \in \mathbb{R}^{d_h}$ from a GNN encoder, as well as its neighbors' embeddings $\mathbf{h}_u$, $\forall u \in \mathcal{N}_v$ where $\mathcal{N}_v$ denotes the neighbor set of $v$. The node embeddings will be fed into the diffusion model to learn the neighbor (positive) distribution of the query node. In the forward diffusion process, we gradually add bite-sized noise to the neighbor node $u$'s embedding. Following the reparameterization trick [15, 28], we can obtain the closed form of the output $\mathbf{h}_{u,t}$ at an arbitrary time step $t$ without relying on the intermediate output:

$$\mathbf{h}_{u,t} = \sqrt{\bar{\alpha}_t}\mathbf{h}_u + \sqrt{1-\bar{\alpha}_t}\epsilon_t, \quad \forall u \in \mathcal{N}_v, \tag{9}$$

In the denoise process, we aim to predict the added noise from the diffused node $u$ at time step $t$ given the query node $v$, which is formulated as

$$\epsilon_{t,\theta|v} = \tau(\mathbf{h}_{u,t}, \mathbf{t}, \mathbf{h}_v; \theta), \quad \forall u \in \mathcal{N}_v, \tag{10}$$

where $\mathbf{t} \in \mathbb{R}^{d_h}$ is a continuous embedding vector for time $t$, and $\theta$ denotes learnable parameters. Following earlier work [15], we employ the sinusoidal positional encoding for the time steps, such that $[\mathbf{t}]_{2i} = \sin(t/10000^{\frac{2i}{d_h}})$ and $[\mathbf{t}]_{2i+1} = \cos(t/10000^{\frac{2i}{d_h}})$, followed by a multilayer perceptron (MLP). $\tau(\cdot; \theta)$ is a learnable transformation function, which estimates the transformation of diffused embeddings at time $t$ to predict the noise. In our model, we implement $\tau$ as a feature-wise linear modulation (FiLM) [34] layer, which is conditioned on both the time step $t$ and query node embedding $\mathbf{h}_v$. Specifically,

$$\epsilon_{t,\theta|v} = (\gamma + \mathbf{1}) \odot \mathbf{h}_{u,t} + \eta, \tag{11}$$

where $\gamma$ and $\eta \in \mathbb{R}^{d_h}$ are scaling and shifting vectors to transform the diffused node embedding, respectively; $\mathbf{1}$ denotes a vector of ones to center the scaling around one, and $\odot$ denotes element-wise multiplication. The transformation vectors are learnable, conditioned on the diffusion time embedding $\mathbf{t}$ and the query node embedding $\mathbf{h}_v$. We implement $\gamma$ and $\beta$ as fully connected layers (FCLs), dependent upon the conditional embeddings as follows.

$$\gamma = \text{FCL}(\mathbf{t} + \mathbf{h}_v; \theta_\gamma), \quad \eta = \text{FCL}(\mathbf{t} + \mathbf{h}_v; \theta_\eta), \tag{12}$$

where $\theta_\gamma, \theta_\eta$ are parameters of the FCLs. We aggregate the time step and query node information by summation, yet other methods such as a neural network can also be considered. Also note that in practical implementations, multiple FiLM layers can be stacked to enhance model capacity.

### 4.2 Training Objective

The diffusion model and GNN encoder are trained simultaneously in an alternating manner. In the outer loop, the GNN encoder is

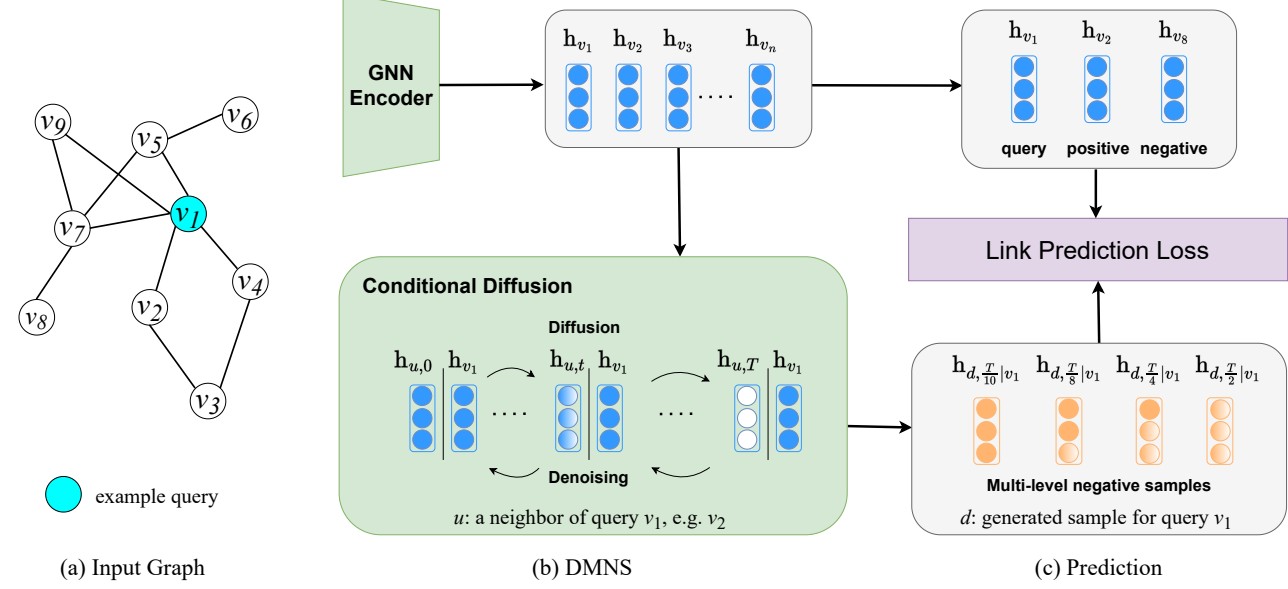

Figure 1: Overall framework of DMNS.

updated for link prediction, taking into account the multi-level negative nodes generated by the diffusion model conditioned on the query node. In the inner loop, the diffusion model is updated to generate the positive neighbor distribution of the query node, based on the current GNN encoder. More details of the training process are outlined in Appendix A. In the following, we discuss the loss functions for the diffusion and GNN-based link prediction.

*Diffusion loss.* We employ a mean squared error (MSE) between sampled noises from the forward process and predicted noises from the reverse process [15]. That is, at time step $t$,

$$\mathcal{L}_D = \|\epsilon_t - \epsilon_{t,\theta|v}\|^2 \quad (13)$$

After the diffusion model is updated, we utilize it to synthesize node embeddings for the main link prediction task. Starting from complete noises ($t = T$), we obtain generated samples by sequentially removing predicted noises at each time step.

$$\mathbf{h}_{d,T|v} \sim \mathcal{N}(\mathbf{0}, \mathbf{I}), \quad (14)$$

$$\mathbf{h}_{d,t-1|v} = \frac{1}{\sqrt{\alpha_t}} \left( \mathbf{h}_{d,t|v} - \frac{1-\alpha_t}{\sqrt{1-\bar{\alpha}_t}} \epsilon_{t,\theta|v} \right) + \sigma_t \mathbf{z}, \quad (15)$$

where the standard deviation $\sigma_t = \sqrt{\beta_t}$ and $\mathbf{z} \sim \mathcal{N}(\mathbf{0}, \mathbf{I})$.

The sequence of generated samples $\{\mathbf{h}_{d,t|v} : 0 \leq t \leq T\}$ represents multi-level negative nodes in the latent space, w.r.t. a query node $v$. We can easily control the number and hardness requirement of the negative sampling, by choosing certain time steps between 0 and $T$. On the one hand, sampling from too many smaller time steps consumes more memory but brings little diversity. On the other hand, sampling from larger time steps may bring in trivial noises which are easier to distinguish. In our implementation, we balance the multi-level strategy by choosing the output from a range of well-spaced steps to form our generated negative set for the query node $v$: $D_v = \{\mathbf{h}_{d,t|v} : t = \frac{T}{10}, \frac{T}{8}, \frac{T}{4}, \frac{T}{2}\}$, which are both efficient and robust.

*Link prediction loss.* We adopt the log-sigmoid loss [13] for the link prediction task. Consider a quadruplet $(v, u, u', D_v)$, where $v$ is a query node, $u$ is a positive node linked to $v$, $u'$ is an existing negative node randomly sampled from the graph, and $D_v$ is a set of multi-level negative nodes w.r.t. the query $v$, which are latent node embeddings generated from the diffusion model at chosen time steps. Note that we still employ existing nodes from the graph as negative nodes, to complement the samples generated in the latent space. Then, the loss is formulated as

$$\mathcal{L} = -\log \sigma(\mathbf{h}_v^\top \mathbf{h}_u) - \log \sigma(-\mathbf{h}_v^\top \mathbf{h}_{u'})$$
$$- \sum_{d_i \in D_v} w_i \log \sigma(-\mathbf{h}_v^\top \mathbf{h}_{d_i})) \quad (16)$$

where $\sigma(\cdot)$ is the sigmoid activation, and $w_i$ is the weight parameter for each negative example generated at different time steps. The idea is, given different levels of hardness associated with different time steps, the importance of the examples from different steps also varies. While various strategies for $w$ can be applied, we use a simple linearly decayed sequence of weights with the assumption that closer steps (harder examples) are more important. Further investigation on the choice and weighting of samples will be discussed in Sect. 5.

### 4.3 Theoretical Analysis

We present a theoretical analysis to justify the samples generated from our conditional diffusion model. Specifically, a **Sub-linear Positivity Principle** [47] has been established earlier for robust negative sampling on graph data. The principle states that the negative distribution should be *positively* but *sub-linearly correlated* to the positive distribution, which has been shown to be able to balance the trade-off between the embedding objective and expected risk. Here, we show that the negative examples generated from the

diffusion model are in fact drawn from a negative distribution that follows the principle.

THEOREM 1 (SUB-LINEAR POSITIVITY DIFFUSION). *Consider a query node $v$. Let $\mathbf{x}_n \sim \mathcal{N}(\mu_{t,\theta}, \Sigma_{t,\theta})$ and $\mathbf{x}_p \sim \mathcal{N}(\mu_{0,\theta}, \Sigma_{0,\theta})$ represent samples drawn from the negative and positive distributions of node $v$, respectively. Suppose the parameters of the two distributions are specified by a diffusion model $\theta$ conditioned on the query node $v$ at time $t > 0$ and $0$, respectively. Then, the density function of the negative samples $f_n$ is sub-linearly correlated to that of the positive samples $f_p$:*

$$f_n(\mathbf{x}_n | v) \propto f_p(\mathbf{x}_p | v)^{\lambda}, \quad \text{for some } 0 < \lambda < 1, \quad (17)$$

*as long as $\Psi \geq 0$, which is a random variable given by $\Psi = 2\Delta^{\top}\sqrt{\bar{\alpha}_t}(\mathbf{x}_0 - \mu_0) + \Delta^{\top}\Delta \geq 0$, where $\Delta = \sqrt{\bar{\alpha}_t}\mu_0 + \sqrt{1 - \bar{\alpha}_t}\epsilon_0 - \mu_t$, $\mathbf{x}_0$ is generated by the model $\theta$ at time $0$, and $\epsilon_0 \sim \mathcal{N}(\mathbf{0}, \mathbf{I})$.* □

PROOF. Note that conditional diffusion $\theta$ aims to learn neighbor (positive) distribution of query node $v$, the samples generated from time step 0 can be regarded as positive samples while their counterparts from larger $t > 0$ treated as negative ones. Then the density functions of the generated positive and negative sampling distributions are as follows:

$$f_p(\mathbf{x}_p | v) = \mathcal{N}(\mathbf{x}_0; \mu_0, \Sigma_1^2)$$
$$= \frac{1}{2\pi^{k/2} \det(\Sigma_1)^{1/2}} \exp\left\{-\frac{1}{2}(\mathbf{x}_0 - \mu_0)^{\top}\Sigma_1^{-1}(\mathbf{x}_0 - \mu_0)\right\} \quad (18)$$
$$= \frac{1}{2\pi^{k/2}\beta_1^{k/2}} \exp\left\{-\frac{1}{2\beta_1}(\mathbf{x}_0 - \mu_0)^{\top}(\mathbf{x}_0 - \mu_0)\right\}$$

$$f_n(\mathbf{x}_n | v) = \mathcal{N}(\mathbf{x}_t; \mu_t, \Sigma_{t+1}^2)$$
$$= \frac{1}{2\pi^{k/2} \det(\Sigma_{t+1})^{1/2}} \exp\left\{-\frac{1}{2}(\mathbf{x}_t - \mu_t)^{\top}\Sigma_{t+1}^{-1}(\mathbf{x}_t - \mu_t)\right\} \quad (19)$$
$$= \frac{1}{2\pi^{k/2}\beta_{t+1}^{k/2}} \exp\left\{-\frac{1}{2\beta_{t+1}}(\mathbf{x}_t - \mu_t)^{\top}(\mathbf{x}_t - \mu_t)\right\}$$

where $\mu_0 = \mu_{\theta}(\mathbf{x}_1, 1, \mathbf{v}), \mu_t = \mu_{\theta}(\mathbf{x}_{t+1}, t+1, \mathbf{v}), \Sigma_i = \beta_i\mathbf{I}$, $k$ is the vector dimension.

By applying reparameterization trick [15, 28] we derive:

$$\mathbf{x}_t = \sqrt{\bar{\alpha}_t}\mathbf{x}_0 + \sqrt{1 - \bar{\alpha}_t}\epsilon_0 \quad (20)$$

where $\epsilon_0 \sim \mathcal{N}(\mathbf{0}, \mathbf{I})$. Replacing $x_t$ by $x_0$ in $f_n(\mathbf{x}_n | v)$ we obtain:

$$f_n(\mathbf{x}_n | v) = \frac{1}{2\pi^{k/2}\beta_{t+1}^{k/2}} \exp\left\{-\frac{1}{2\beta_{t+1}}(\sqrt{\bar{\alpha}_t}\mathbf{x}_0 + \sqrt{1 - \bar{\alpha}_t}\epsilon_0 - \mu_t)^{\top}\right.$$
$$\left. (\sqrt{\bar{\alpha}_t}\mathbf{x}_0 + \sqrt{1 - \bar{\alpha}_t}\epsilon_0 - \mu_t)\right\} \quad (21)$$

$$= \frac{1}{2\pi^{k/2}\beta_{t+1}^{k/2}} \exp\left\{-\frac{1}{2\beta_{t+1}}\left[\sqrt{\bar{\alpha}_t}(\mathbf{x}_0 - \mu_0) + \Delta\right]^{\top}\right.$$
$$\left. \left[\sqrt{\bar{\alpha}_t}(\mathbf{x}_0 - \mu_0) + \Delta\right]\right\} \quad (22)$$

$$= \frac{1}{2\pi^{k/2}\beta_{t+1}^{k/2}} \exp\left\{-\frac{1}{2\beta_{t+1}}\left[\bar{\alpha}_t(\mathbf{x}_0 - \mu_0)^{\top}(\mathbf{x}_0 - \mu_0)\right.\right.$$
$$\left.\left. +2\Delta^{\top}\sqrt{\bar{\alpha}_t}(\mathbf{x}_0 - \mu_0) + \Delta^{\top}\Delta\right]\right\} \quad (23)$$

with $\Delta = \sqrt{\bar{\alpha}_t}\mu_0 + \sqrt{1 - \bar{\alpha}_t}\epsilon_0 - \mu_t$. We denote $\Psi = 2\Delta^{\top}\sqrt{\bar{\alpha}_t}(\mathbf{x}_0 - \mu_0) + \Delta^{\top}\Delta$. If $\Psi \geq 0$, then:

$$\bar{\alpha}_t(\mathbf{x}_0 - \mu_0)^{\top}(\mathbf{x}_0 - \mu_0) + \Psi \geq \bar{\alpha}_t(\mathbf{x}_0 - \mu_0)^{\top}(\mathbf{x}_0 - \mu_0) \quad (24)$$

$$\equiv -\frac{1}{2\beta_{t+1}}\left[\bar{\alpha}_t(\mathbf{x}_0 - \mu_0)^{\top}(\mathbf{x}_0 - \mu_0) + \Psi\right] \quad (25)$$

$$\leq -\frac{1}{2\beta_{t+1}}\bar{\alpha}_t(\mathbf{x}_0 - \mu_0)^{\top}(\mathbf{x}_0 - \mu_0) \quad (26)$$

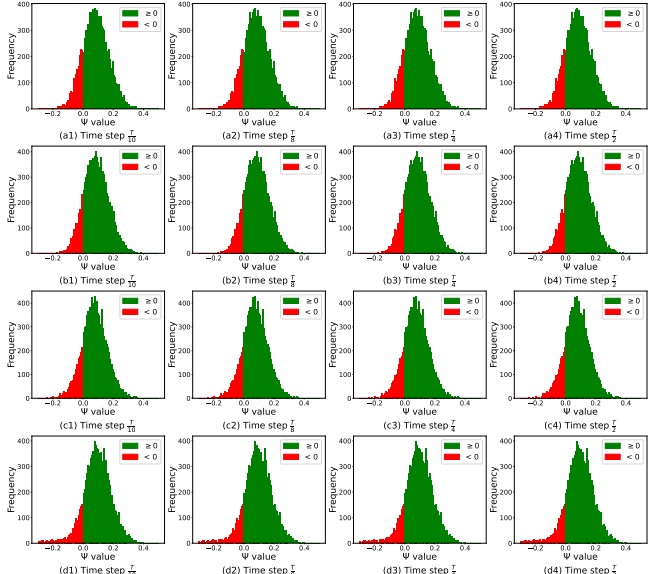

**Figure 2: Empirical distributions (histograms) of $\Psi$ on (a1–a4) Cora, (b1–b4) Citeseer, (c1–c4) Coauthor-CS, (d1–d4) Actor, across different time steps.**

$$\equiv f_n(\mathbf{x}_n | v) \leq \frac{1}{2\pi^{k/2}\beta_{t+1}^{k/2}} \exp\left\{-\frac{1}{2\beta_{t+1}}\bar{\alpha}_t(\mathbf{x}_0 - \mu_0)^{\top}(\mathbf{x}_0 - \mu_0)\right\} \quad (27)$$

$$\leq \frac{\beta_1^{k/2}}{\beta_{t+1}^{k/2}} \frac{1}{2\pi^{k/2}\beta_1^{k/2}} \exp\left\{-\frac{\beta_1}{2\beta_1\beta_{t+1}}\bar{\alpha}_t(\mathbf{x}_0 - \mu_0)^{\top}(\mathbf{x}_0 - \mu_0)\right\} \quad (28)$$

$$\leq \frac{\beta_1^{k/2}}{\beta_{t+1}^{k/2}} \frac{1}{2\pi^{k/2}\beta_1^{k/2}} \exp\left\{-\frac{1}{2\beta_1}(\mathbf{x}_0 - \mu_0)^{\top}(\mathbf{x}_0 - \mu_0)\right\}^{\lambda} \quad (29)$$

$$\propto f_p(\mathbf{x}_p | v)^{\lambda} \quad (30)$$

where $0 < \lambda = \left(\frac{\beta_1}{\beta_{t+1}}\right)\bar{\alpha}_t < 1$ ($\beta_1 < \beta_{t+1}$ through variances scheduling and $0 < \bar{\alpha}_t < 1$). Therefore, the density function of negative samples is sub-linearly correlated to that of positive samples under the constraint $\Psi = 2\Delta^{\top}\sqrt{\bar{\alpha}_t}(x_0 - \mu_0) + \Delta^{\top}\Delta \geq 0$. □

Specifically, we run an experiment to verify the distribution of $\Psi$. We use our diffusion model to generate a large number of examples at time 0 and a given $t$, to compute their mean $\mu_0$ and $\mu_t$, respectively. We present the empirical distributions (histograms) of $\Psi$ on the four datasets, across time steps $\frac{T}{10}, \frac{T}{8}, \frac{T}{4}, \frac{T}{2}$ in Fig. 2. The probabilities that $\Psi \geq 0$ averaged over samples from 4 time steps on Cora, Citeseer, Coauthor-CS and Actor are 80.14%, 81.62%, 81.65% and 84.31%, respectively. The results indicate that the majority of generated examples from DMNS adhere to the sub-linear positivity theorem in practice.

## 4.4 Algorithm and Complexity

We outline the model training for DMNS in Algorithm 1. In line 1, we initialize the model parameters. In line 3, we sample a batch of triplets from training data. In lines 4–6, we obtain embeddings for all nodes in training batch by the GNN encoder. In lines 8–16, we train

---

**Algorithm 1** MODEL TRAINING FOR DMNS

**Input:** Graph $G = (V, E)$, training triplets $T = (v, u, u')$
**Output:** GNN model parameters $\omega$, Diffusion model parameters $\theta$.
1: initialize parameters $\omega$, $\theta$
2: **while** not converged **do**
3:     sample a batch of triplets $T_b \subset T$;
4:     **for** each node $v$ in the batch $T_b$ **do**
5:         $\mathbf{h}_v = \sigma \left( \text{AGGR}(\mathbf{h}_v, \{\mathbf{h}_i : i \in \mathcal{N}_v\}; \omega) \right)$;
6:     //Train diffusion
7:     **while** not converged **do**
8:         **for** each query node $v$ in the batch $T_b$ **do**
9:             $u \sim N_v$,  $\epsilon \sim \mathcal{N}(0, \mathbf{I})$;
10:           $t \sim Uniform(1, .., T)$,  $\mathbf{t} = MLP(t)$;
11:           $\epsilon_{t,\theta|v} = \tau(\mathbf{h}_{u,t}, \mathbf{t}, \mathbf{h}_v; \theta)$;
12:           $\mathcal{L}_D = \|\epsilon_t - \epsilon_{t,\theta|v}\|^2$;
13:         update $\theta$ by minimizing $\mathcal{L}_D$ with $\omega$ fixed;
14:     //Sampling multi-level generated negative nodes
15:     **for** each query node $v$ in the batch $T_b$ **do**
16:         $\mathbf{h}_{d,T|v} \sim \mathcal{N}(0, \mathbf{I})$,
17:         **for** $t$ = T-1,.., 0 **do**
18:            $\mathbf{z} \sim \mathcal{N}(0, \mathbf{I})$ if $t > 1$ else $\mathbf{z} = 0$ ;
19:            $\mathbf{h}_{d,t-1|v} = \frac{1}{\sqrt{\alpha_t}} \left( \mathbf{h}_{d,t|v} - \frac{1-\alpha_t}{\sqrt{1-\bar{\alpha}_t}} \epsilon_{t,\theta|v} \right) + \sigma_t \mathbf{z}$;
20:         $D_v = \{\mathbf{h}_{d,t|v} : t = \frac{T}{10}, \frac{T}{8}, \frac{T}{4}, \frac{T}{2}\}$
21:     Calculate $\mathcal{L}$ as Eqn. (15);
22:     update $\omega$ by minimizing $\mathcal{L}$ with $\theta$ fixed;
23: **return** $\omega$, $\theta$.

---

conditional diffusion model to learn the neighborhood distribution of given query node $v$. Specifically, we calculate the predicted noise at arbitrary time step $t$ conditioned on query node in line 12. We compute the diffusion loss and update diffusion parameters in lines 14–15. In lines 18–25, we sample a set multi-level negative nodes for query node by diffusion model. In lines 26-27, we compute the main link prediction loss and update the parameters of GNN encoder.

We analyze the complexity of DMNS for one node. Taking GCN as base encoder, the neighborhood aggregation for one node in the $l$-th layer has complexity $O(d_l d_{l-1} \bar{d})$, where $d_l$ is the dimension of the $l$-th layer and $\bar{d}$ is the average node degree. The computation for diffusion model includes time embeddings module and noise estimation module. The time embeddings module employs a MLP of $L_1$ layers, where a $l$-layer has complexity $O(d_l d_{l-1})$, with $d_l$ is the layer dimension. The noise estimation module employs a neural network of $L_2$ FiLM layers, where $l$-th layer has complexity of $O(2d_l d_{l-1})$ with $d_l$ is the layer dimension. Thus, the diffusion training takes $N$ iterations has complexity $O(NL_1 L_2 d_l d_{l-1})$. After that, the sampling runs the reverse process of $T$ steps (lines 20–23) has complexity $O(2T d_l d_{l-1})$. Compared to the base GCN, the overhead of diffusion part has complexity $O\left((2T + NL_1 L_2) d_l d_{l-1}\right)$.

## 5 EXPERIMENTS

In this section, we conduct extensive experiments to evaluate the effectiveness of DMNS [1] on several benchmark datasets, and analyze several key aspects of the model.

---

[1]Code & data at https://github.com/Anonymous235876/DMNS.git for review.

---

**Table 1: Summary of datasets.**

| Datasets | Nodes | Edges | Features | Property |
|----------|-------|-------|----------|----------|
| Cora | 2708 | 5429 | 1433 | homophilous |
| Citeseer | 3327 | 4732 | 3703 | homophilous |
| Coauthor-CS | 18333 | 163788 | 6805 | homophilous |
| Actor | 7600 | 30019 | 932 | heterophilous |

### 5.1 Experimental Setup

*Datasets.* We employ four public graph datasets, summarized in Table 1. Cora and Citeseer [46] are two citation networks, where each node is a document and the edges represent citation links. Coauthor-CS [36] is a co-authorship network, where each node is an author and an edge exists if they co-authored a paper. Actor [33] is an actor co-occurrence network, where each node denotes an actor and each edge connects two actors both occurring on the same Wikipedia page.

*Baselines.* We employ a comprehensive suite of baselines for the link prediction task. (1) *Classical GNNs*: GCN [22] and VGAE [21], which are classical GNNs models for link prediction. (2) *Heuristic negative sampling*: PNS [29], DNS [51] and MCNS [47], which employ various heuristics to retrieve hard negative examples. (3) *Generative adversarial methods*: GraphGAN [43], ARVGA [32] and KBGAN [4], which leverages GANs to learn the negative distribution and select hard examples. (4) *Subgraph-based GNNs*: SEAL [50] and ScaLed [26], which utilizes local subgraphs surrounding the candidate nodes. See Appendix A for a more detailed description of the baselines.

*Task setup and evaluation.* On each graph, we randomly split its links for training, validation and testing following the proportions 90%:5%:5%. Note that the graphs used in training are reconstructed from only the training links. We adopt a ranking-based link prediction during testing. Given a query node $v$, we sample a positive candidate $u$ such that $(v, u)$ is a link in the test set, and further sample 9 nodes that are not linked to $v$ as negative candidates. For evaluation, we rank the 10 candidate nodes based on their dot product with the query node $v$. Based on the ranked list, we report two ranking-based metrics, namely, NDCG and MAP [25], averaged over five runs.

*Parameters and settings.* For our model DMNS, by default, we employ GCN [22] as the base encoder. To further evaluate the effectiveness of DMNS on different encoders, we also conduct experiments using GAT [42] and GraphSAGE (SAGE) [13]. For GCN, we employ two layers with dimension 32. For the diffusion model, we use two FiLM layers with output dimension 32, set the total time step as $T = 50$, and assign the variances $\beta$ as constants increasing linearly from $10^{-4}$ to 0.02. We set the weights of negative examples to $\{1, 0.9, 0.8, 0.7\}$ in Eq. (16), corresponding to time steps $\{\frac{T}{10}, \frac{T}{8}, \frac{T}{4}, \frac{T}{2}\}$. For all baselines, we adopt the same GNN architecture and settings as in DMNS for a fair comparison. They also employ a log-sigmoid objective [13] consistent with our link prediction loss. For those baselines which do not propose a negative sampling method, we perform the uniform negative sampling if

**Table 2: Evaluation of link prediction against baselines using GCN as the base encoder.**

| Methods | Cora | | Citeseer | | Coauthor-CS | | Actor | |
|---|---|---|---|---|---|---|---|---|
| | MAP | NDCG | MAP | NDCG | MAP | NDCG | MAP | NDCG |
| GCN | .742 ± .003 | .805 ± .003 | .735 ± .011 | .799 ± .008 | .823 ± .004 | .867 ± .003 | .521 ± .004 | .634 ± .003 |
| GVAE | .783 ± .003 | .835 ± .002 | .743 ± .004 | .805 ± .003 | .843 ± .011 | .882 ± .008 | .587 ± .004 | .684 ± .003 |
| PNS | .730 ± .008 | .795 ± .006 | .748 ± .006 | .809 ± .005 | .817 ± .004 | .863 ± .003 | .517 ± .006 | .631 ± .006 |
| DNS | .735 ± .007 | .799 ± .005 | .777 ± .005 | .831 ± .004 | .845 ± .003 | .883 ± .002 | .558 ± .006 | .663 ± .005 |
| MCNS | .756 ± .004 | .815 ± .003 | .750 ± .006 | .810 ± .004 | .824 ± .004 | .868 ± .004 | .555 ± .005 | .659 ± .004 |
| GraphGAN | .739 ± .003 | .802 ± .002 | .740 ± .011 | .803 ± .008 | .818 ± .007 | .863 ± .005 | .534 ± .007 | .644 ± .005 |
| ARVGA | .732 ± .011 | .797 ± .009 | .689 ± .005 | .763 ± .004 | .811 ± .003 | .858 ± .002 | .526 ± .012 | .638 ± .009 |
| KBGAN | .615 ± .004 | .705 ± .003 | .568 ± .006 | .668 ± .005 | .852 ± .002 | .888 ± .002 | .472 ± .003 | .596 ± .002 |
| SEAL | .751 ± .007 | .812 ± .005 | .718 ± .002 | .784 ± .002 | .850 ± .001 | .886 ± .001 | .536 ± .001 | .641 ± .001 |
| ScaLed | .676 ± .004 | .752 ± .003 | .630 ± .004 | .712 ± .003 | .828 ± .001 | .869 ± .001 | .459 ± .001 | .558 ± .001 |
| DMNS | **.793** ± .003 | **.844** ± .002 | **.790** ± .004 | **.841** ± .003 | **.871** ± .002 | **.903** ± .001 | **.600** ± .002 | **.696** ± .002 |

*Best is **bolded** and runner-up underlined.

**Table 3: Evaluation of link prediction on DMNS with various base encoders.**

| Methods | Cora | | Citeseer | | Coauthor-CS | | Actor | |
|---|---|---|---|---|---|---|---|---|
| | MAP | NDCG | MAP | NDCG | MAP | NDCG | MAP | NDCG |
| GAT | .766 ± .006 | .824 ± .004 | .767 ± .007 | .763 ± .062 | .833 ± .003 | .874 ± .002 | .479 ± .004 | .603 ± .003 |
| DMNS-GAT | **.813** ± .004 | **.859** ± .003 | **.788** ± .007 | **.840** ± .006 | **.851** ± .002 | **.889** ± .002 | **.573** ± .007 | **.675** ± .005 |
| SAGE | .598 ± .014 | .668 ± .013 | .622 ± .012 | .713 ± .009 | .768 ± .005 | .826 ± .004 | .486 ± .004 | .604 ± .003 |
| DMNS-SAGE | **.700** ± .007 | **.773** ± .005 | **.669** ± .013 | **.749** ± .010 | **.843** ± .004 | **.883** ± .003 | **.582** ± .017 | **.682** ± .013 |

required for training. Additional settings of our method and the baselines can be found in Appendix B.

## 5.2 Evaluation on Link Prediction

We evaluate the performance of DMNS on link prediction against various baselines and with different base encoders.

*Comparison with baselines.* We report the results of DMNS and the baselines in Table 2. Overall, DMNS significantly outperforms competing baselines across all datasets and metrics. The results indicate that our strategy of multi-level negative sampling is effective. We make three further observations. First, heuristic negative sampling methods generally improve the performance upon the base encoder with uniform sampling (*i.e.*, GCN), showing the utility of harder examples. Among them, DNS and MCNS are more robust as they leverage more sophisticated heuristics to select better negative examples. Second, GAN-based methods are often worse than classical GNNs, which is potentially due to the difficulty in training GANs. For instance, GVAE achieves a relatively competitive performance, while ARVGA—its variant with an adversarial regularizer—suffers from a considerable drop. Third, subgraph-based GNNs achieve mixed results, suggesting that the local structures can be effective to some extent yet high-quality negative examples are still needed for training.

*Evaluation with other encoders.* We further utilize GAT and SAGE as the base encoders, and the corresponding model DMNS-GAT and DMNS-SAGE, respectively. From the results reported in Table 3, we observe that DMNS achieves significant improvements compared

to its corresponding base encoders in all cases, which demonstrates the flexibility of DMNS to effectively work with various encoders.

## 5.3 Model Analyses

Finally, we investigate various aspects of DMNS through ablation studies, parameter sensitivity analysis and visualization on the four datasets.

*Ablation studies.* The first study investigates the ablation on model design, to demonstrate the effectiveness of each module in DMNS. We compare with two variants: (1) *unconditional* diffusion, by removing the condition on the query node (and its associated neighboring nodes) such that the diffusion model now generates arbitrary node embeddings from noises; (2) *unweighted* negative examples, by setting all $w_i = 1$ in Eq. (16). We report the results in Fig. 3, and make the following observations. First, unconditional diffusion performs significantly worse than its conditional counterpart, since the generated negative examples are not optimized for a specific query node. Second, the unweighted variant also exhibits a drop in performance, implying that negative examples from different levels of hardness have different importance.

The second study involves the ablation on the sampling choice. Instead of sampling a mixture of examples from well-spaced time steps, we now only use output from a single time step for each query node. We vary the time step between $\frac{T}{10}$ and $\frac{T}{2}$, as shown in Fig. 4. The performance of each single time step varies but all are worse than combining them together (*i.e.*, DMNS), demonstrating the effectiveness of multi-level sampling. It is also observed that

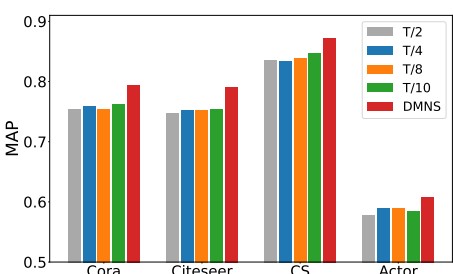

**Figure 3: Ablation on model design.**

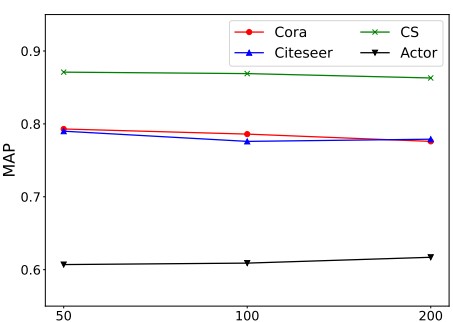

**Figure 4: Ablation on sampling choices.**

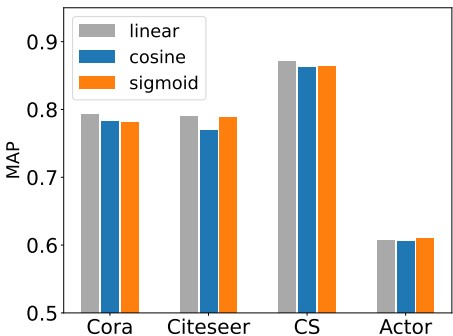

**Figure 5: Impact of total timestep $T$.**

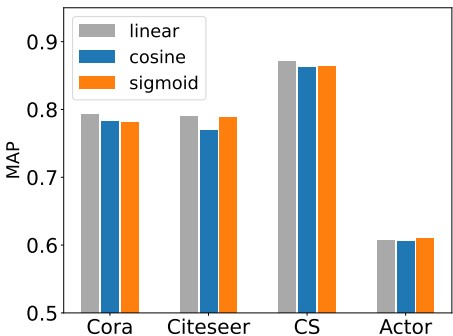

**Figure 6: Impact of variances $\beta$.**

smaller time steps ($\frac{T}{10}$, $\frac{T}{8}$) often outperform larger time steps ($\frac{T}{4}$, $\frac{T}{2}$), indicating that harder examples could be more useful.

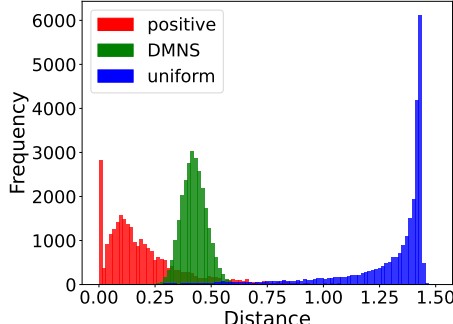

**Figure 7: Histogram of embedding distances from query.**

*Parameter studies.* We showcase the impact of two important hyperparameters, including the total time step $T$ for the diffusion process, and the scheduling for diffusion variances $\beta$. In Fig. 5, we observe that increasing total time steps may not boost the performance while requiring more computation. In Fig. 6, results show that linear scheduling achieves the best performance among several standard scheduling policies including cosine and sigmoid scheduling [31].

*Visualization.* We further investigate the quality of the negative examples generated by DMNS. For each query node, we construct a positive set consisting of its neighbors, a negative set consisting of negative examples from DMNS at time $\frac{T}{10}$, and a second negative set consisting of uniformly sampled nodes from the graph. We calculate the Euclidean distance between the query node embedding and the embeddings in each set, and plot the empirical distribution (histogram) of the distances in Fig. 7. For the positive set, we naturally expect smaller distances to the query node. For the negative sets, the distance can be regarded as a proxy to hardness: Smaller distances from the query node imply harder examples. As we can see, DMNS produces harder examples than uniform sampling, but are not too hard (*i.e.*, not closer to the query node than the positives) to impair the performance [9, 44].

## 6 CONCLUSION

In this paper, we investigated a novel strategy of multi-level negative sampling for graph link prediction. Existing methods aim to retrieve hard examples heuristically or adversarially, but they are often inflexible or difficult to control the "hardness". In response, we proposed a novel sampling method named DMNS based on a conditional diffusion model, which empowers the sampling of negative examples at different levels of hardness. In particular, the hardness can be easily controlled by sampling from different time steps of the denoise process within the diffusion model. Moreover, we showed that DMNS largely obeys the sub-linear positivity principle for robust negative sampling. Finally, we conducted extensive experiments to demonstrate the effectiveness of DMNS. A limitation of our work is the focus on the effectiveness and robustness of negative sampling for link prediction. Hence, one promising future direction is to optimize the sampling process for efficiency on large-scale graphs. Alternatively, we can investigate the potential of diffusion model in other graph learning tasks such as node classification.

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

# APPENDICES

## A  Details of Baselines

In this section, we describe each baseline in more details.

- *Classical GNNs*: GCN [22] applies mean-pooling aggregates information for the target node by over its neighbors. GVAE [21] proposes an unsupervised training for graph representation learning by reconstructing graph structure.
- *Heuristic negative sampling*: PNS is adopted from word2vec [29], where the distribution for negative samples is calculated by the normalized node degree. DNS [51] dynamically select hard negative examples from candidates ranked by the current prediction scores. MCNS [47] proposes a Metropolis-Hasting method for negative sampling based on the sub-linear positivity principle.
- *Generative adversarial methods*: GraphGAN [43] samples hard negative examples from the learned connectivity distribution through adversarial training. KBGAN [4] samples high quality negative examples for knowledge graph embeddings by the generator that learns to produce distribution for negative candidates conditioned on the positives. ARVGA [32] integrates an adversarial regularizer to GVAE to enforce the model on generating realistic samples.
- *Subgraph-based GNNs*: SEAL [50] proposes to sample k-hops local subgraphs surrounding two candidates for link prediction. ScaLED [26] improves SEAL by utilizing random walks for efficient subgraphs sampling.

For other base GNN encoders, GAT [42] employs self-attention mechanism to produce learnable weights to each neighbor of target node during aggregation. SAGE [13] first aggregates information from target node neighbors, then concatenates with the node itself to obtain the node embedding.

## B  Model Settings

For all the approaches, we use a two layers GCN as encoder with output dimension as 32 for fair comparison to conduct link prediction. For our model DMNS, we set learning rate as 0.01, dropout ratio as 0.1. For each query nodes $v$, we sample a neighbor set $N_v = 20$ for diffusion training.

For GCN, we set the hidden dimension as 32. For GAT, we use three attention heads with hidden dimension for each head as 32. For SAGE, we use mean aggregator for concatenation and set its hidden dimension as 32. In PNS, we utilize node degree to calculate the popularity distribution and normalize it to $3/4^{th}$ power. For DNS, we set the number of negative candidates as 10 and select 1 negative sample as the highest ranked node. For MCNS, we set the DFS length as 5 and the proposal distribution $q(y|x)$ is mixed betwwen uniform and k-nearest nodes sampling wih $p = 0.5$. In these methods we set learning rate as 0.01.

For GraphGAN, we use pretrained GCN to obtain initialized node embeddings and set the learning rates for discriminator $D$ and generator $G$ as 0.0001. In each iteration, we set the number of gradient updates for both $D$ and $G$ as 1 for best results. For KBGAN, we adapt it to the homogeneous graph setting by neglecting all relation types and use a single embedding to represent a universal relation. We further set the number of negative examples as 20 and number of gradient updates for both $D$ and $G$ as 1. For ARVGA, learning rates for encoder $E$ and discriminator $D$ are 0.005 and 0.001 respectively, hidden dimension for $D$ is 64, the number of gradient updates $D$ for each iteration is 5.

For Subgraph-based GNNs approaches, we replace their original objective Binary Cross Entropy with our Log Sigmoid and do not utilize additional node structural embeddings for fair evaluation. For SEAL, we set learning rate as 0.0001, the number of hop $k$ equals 1. On ScaLed, we set random walk length as 3 and number of walks as 20.

## C  Environment

All experiments are conducted on a workstation with a 12-core CPU, 128GB RAM, and 2 RTX-A5000 GPUs. We implemented DMNS using Python 3.8 and Pytorch 1.13 on Ubuntu-20.04.

