# OpenReview forum: "Diffusion-based Negative Sampling on Graphs for Link Prediction"
_ACM.org/TheWebConf/2024/Conference — TheWebConf24 Oral_

### Official Review · Reviewer_YZSA · 2023-11-21

**Novelty:** 5
**Technical Quality:** 5

**Review:**

# Overall Summary
This paper focuses on the graph representation learning problem. Existing learning models typically follow a contrastive learning manner, where negative sampling is pivotal. Existing negative sampling methods are either heuristic or inflexible to control. To address this issue, the authors propose a multi-level negative sampling strategy, where the difficulty level of negative samples is controlled via sampling from different time steps of the diffusion process. The authors theoretically show that the proposed method follows the sub-linear positivity principle for robust negative sampling and empirically show that the proposed method can achieve superior performance on various datasets.

# Strengths
- The idea of multi-level negative samples is reasonable and the implementation based on the diffusion model is novel. Controlling the difficulty level of the negative samples by sampling from different time steps is reasonable and the results are promising.
- The performance gain is significant on the link prediction task.
- The paper is well-organized and easy to follow.

# Weaknesses
- The setting of link prediction limits the applicability of the proposed method. The overall method is a general graph learning method and the implementation does not rely too much on the task setting of link prediction. Therefore, the authors should also verify the method's effectiveness on the node classification task. Though the authors leave this as future works, I think that it will be better to include this part in the current version.
- The theoretical analysis is different from the original claim in ref. [47]. Ref. [47] presents the sub-linear positivity principle as $f_n(u|v) \propto f_p(u|v)^\lambda$, while the authors prove that $f_n(u'|v) \propto f_p(u|v)^\lambda$ where $u'$ is a negative sample. I doubt whether these two claims are equivalent. Moreover, the meaning of the introduced term $\Psi$ has not been explained.
- The experiment part can be improved.
    - The negative samples for training comprise heuristic samples and generated samples. The ablation study should verify the effectiveness of these two sample sets.
    - Empirical time analysis should be included.
    - Figure 7 should include more negative samples at different time steps.

# Conclusion
This paper is a borderline paper. The motivation and the model design are reasonable and promising, while the theoretical analysis and the experimental design need improving.

# Rebuttal
I have read the rebuttal.

**Questions:**

Please refer to the weaknesses section.

**Reviewer Confidence:**

3: The reviewer is confident but not certain that the evaluation is correct

**Scope:**

4: The work is relevant to the Web and to the track, and is of broad interest to the community

---

### Official Review · Reviewer_LutX · 2023-11-21

**Novelty:** 6
**Technical Quality:** 6

**Review:**

This paper proposes a novel strategy of multi-level negative sampling for graph link prediction. It proposes a novel sampling method named DMNS based on a conditional diffusion model, which empowers the sampling of negative examples at different levels of hardness. They present a theoretical analysis and show that DMNS obeys the sub-linear positivity principle for robust negative sampling. Extensive experiments are conducted to demonstrate the effectiveness of DMNS.

Pros:
1. The idea of leveraging the Markov chain property of diffusion models to generate negative nodes in multiple levels of variable hardness for graph link prediction is interesting.
2. The paper is well-written and easy to follow.
3. Theoretical analysis is provided to show the sub-linear positivity principle of DMNS.
Cons:
1. The sizes of the datasets are mostly small, is DMNS scalable to large-scale graphs? If not, why? If so, more datasets should be evaluated.
2. Only theoretical complexity analysis is provided, the authors should conduct experimental analysis on both time and space complexity.

**Questions:**

1. Is DMNS scalable to large-scale graphs?
2. Experimental analysis on both time and space complexity should be provided.

**Reviewer Confidence:**

2: The reviewer is willing to defend the evaluation, but it is likely that the reviewer did not understand parts of the paper

**Scope:**

4: The work is relevant to the Web and to the track, and is of broad interest to the community

---

### Official Review · Reviewer_g2S9 · 2023-11-23

**Novelty:** 6
**Technical Quality:** 6

**Review:**

## Summary
This paper motivates that the existing negative sampling strategy for training graph representation learning based link prediction models is inefficient in controlling the "hardness" of the generated negative samples. To address this issue, this paper proposes to learn a diffusion model based negative sampling approach to generate for a given node negative samples that interpolates from a noise node representation and the representation of a 1-hop neighbor. The diffusion's time step thus indicates the "hardness" of the negative sample. The diffusion model is trained together with the representation learning model in an alternating manner. Empirical study shows the proposed approach effectively improved the link prediction results for GCN, GAT and GraphSAGE. The paper theoretically shows that the proposed diffusion model generates negative samples follows the sublinear positivity principle, though I did not check the proof in detail. Ablation study as well as parameter analysis are also provided to provide empirical evidence supporting the design choices of the paper.

## Strong points
* The idea of train a diffusion model conditioned on query nodes to generate negative samples is novel in my opinion.
* The design of the diffusion model architecture, as well as the training process and loss function are sound.
* The empirical results supports the effectiveness claim of the proposed approach.
* The paper is easy to follow.

## Suggestions
* The motivating limitation of controlling "hardness"  in the existing approach could further be strengthened by qualitative analysis or via a toy example. Through such analysis, the claim that more controlled negative sample could result in a better representation model hence link prediction results, would be come stronger and more intuitive.
* The paper mentioned one the limitation is the effectiveness of negative sampling, it would be great to briefly discuss empirically to which magnitude the training time would be increased. This would also give the practitioners a good idea for trying out the proposed approach.

**Questions:**

See "suggestions"

**Reviewer Confidence:**

3: The reviewer is confident but not certain that the evaluation is correct

**Scope:**

4: The work is relevant to the Web and to the track, and is of broad interest to the community

---

### Official Review · Reviewer_Y6fZ · 2023-11-27

**Novelty:** 3
**Technical Quality:** 4

**Review:**

Summary: This paper proposes a negative sampling method for graph link prediction.  This paper further considers the optimal samples in the latent space compared with the classical methods. The proposed method uses the Markov chain property of diffusion models to generate negative nodes.

Strengths:
(1) This paper is not hard to follow. The motivation mentioned in the Introduction is clear.
(2) In Section 4, the authors provide a clear theoretical analysis of their proposed method. This can help readers follow their core ideas.
(3) The experimental setting is reasonable and the result seems good.

Weaknesses:
(1) This paper ignores some highly relevant works, such as "Efficient Link Prediction via GNN Layers Induced by Negative Sampling" and "Negative sampling and rule mining for explainable link prediction in knowledge graphs". Thus, it is difficult to convince me that the authors have done a fair comparison in the experiment.
(2) As for the technical details, compared with the existing models, the contribution of this paper is incremental. The proposed method is just a mix of the classical link prediction methods with the existing denoising diffusion probabilistic models.

**Questions:**

1. Why did not compare the proposed method with some novel baselines of negative sampling-based link prediction?
2. Why not provide the rationale behind the proposed DMNS? The authors should highlight the motivation for technical improvements.

**Reviewer Confidence:**

3: The reviewer is confident but not certain that the evaluation is correct

**Scope:**

4: The work is relevant to the Web and to the track, and is of broad interest to the community

---

### Decision · Program_Chairs · 2024-01-22

**Decision:**

Accept (Oral)

**Comment:**

**Meta Review**: This paper proposes a strategy of multi-level negative sampling for graph link prediction. It proposes a novel sampling method named DMNS based on a conditional diffusion model, which empowers the sampling of negative examples at different levels of hardness. This paper was generally enjoyed by reviewers (indeed most reviewers could not find many weaknesses to comment on). In my opinion the discussion has a strong consensus on accept.

 **Strengths**
 + novelty (g2S9, LutX, YZSA)
 + empirical results (g2S9, YZSA)
 + readability (g2S9, LutX, YZSA)
 + theory (LutX)

 **Weaknesses**
 - scalability (LutX)
 - method is actually more useful than just link prediction (YZSA)
 - additional baselines would be interesting (Y6fZ)